# Oilseed Rape Shares Abundant and Generalized Pollinators with Its Co-Flowering Plant Species

**DOI:** 10.3390/insects12121096

**Published:** 2021-12-08

**Authors:** Amibeth Thompson, Valentin Ștefan, Tiffany M. Knight

**Affiliations:** 1Institute of Biology, Martin Luther University Halle-Wittenberg, Am Kirchtor 1, 06108 Halle (Saale), Germany; valentin.stefan@idiv.de (V.Ș.); tiffany.knight@idiv.de (T.M.K.); 2German Centre for Integrative Biodiversity Research (iDiv) Halle-Jena-Leipzig, Puschstraße 4, 04103 Leipzig, Germany; 3Department of Community Ecology, Helmholtz Centre for Environmental Research—UFZ, Theodor-Lieser-Straße 4, 06120 Halle (Saale), Germany

**Keywords:** oilseed rape, community composition, floral functional traits, null model, plant-pollinator network, Bray-Curtis index, modularity

## Abstract

**Simple Summary:**

Plants in semi-natural areas provide food resources for pollinators that visit pollinator-dependent crop species, such as Oilseed Rape (OSR). Here, we study the patterns of pollinator visitation on OSR and its co-flowering plants in adjacent semi-natural areas. We find that OSR is visited by pollinators that are abundant in the community and that these pollinators also visit co-flowering plant species in semi-natural areas. OSR primarily influences the pollination of plant species which have similar floral traits (i.e., other disc flowers). Plant species that attract a high abundances of bumblebees, wild bees, flies, and beetles influence the pollination of OSR the most. Our results suggest that plant species in semi-natural areas that support the high abundances of common pollinators which are generalized in their visitation are most important to the pollination of OSR, and that such plant species do not necessarily have similar floral traits to OSR.

**Abstract:**

Mass-flowering crops, such as Oilseed Rape (OSR), provide resources for pollinators and benefit from pollination services. Studies that observe the community of interactions between plants and pollinators are critical to understanding the resource needs of pollinators. We observed pollinators on OSR and wild plants in adjacent semi-natural areas in Sachsen-Anhalt, Germany to quantify (1) the co-flowering plants that share pollinators with OSR, (2) the identity and functional traits of plants and pollinators in the network module of OSR, and (3) the identity of the plants and pollinators that act as network connectors and hubs. We found that four common plants share a high percentage of their pollinators with OSR. OSR and these plants all attract abundant pollinators in the community, and the patterns of sharing were not more than would be expected by chance sampling. OSR acts as a module hub, and primarily influences the other plants in its module that have similar functional traits. However, the plants that most influence the pollination of OSR have different functional traits and are part of different modules. Our study demonstrates that supporting the pollination of OSR requires the presence of semi-natural areas with plants that can support a high abundances of generalist pollinators.

## 1. Introduction

Pollinators underpin food production, since they provide services for approximately 35% of global crop production [1]. While honeybees are traditionally thought of as being the most important pollinating agents, wild bees alone can provide the full pollination requirements of many crops [2,3,4] and the stability of crop pollination increases with the bee richness [2,5,6]. Non-bee insects also contribute a substantial amount to global crop pollination [7]. Wild pollinators’ economical contribution towards crop production is similar to that of honeybees [8]. The abundance and diversity of these wild pollinators, and the quality of the services they provide to crops within agricultural landscapes, is influenced by the composition and quality of the surrounding landscape [3,9]. In order to continue meeting the agricultural demands that come with a growing human population, it is increasingly essential to investigate the factors that may influence pollinator abundance and diversity, and subsequently impact their service to crops [1,10].

Semi-natural areas which surround agricultural fields are important to wild pollinators because they offer a diversity of shelter and nesting sites [11,12] that are not always readily available in agricultural landscapes, and they provide more consistent or diverse food resources for pollinators [9,13,14,15,16]. Mass flowering crops provide a large, but homogenous, food resource to pollinators, which occurs in pulses [17]. Semi-natural areas offer more consistent floral resources over a longtime period [18]. There is a recognition of the importance of semi-natural areas, and practitioners aim to increase floral resources for pollinators with active management (e.g., planting hedgerows near agricultural landscapes). However, the choice of plants in these management activities is often based on pollinator syndromes, rather than on the ecological observations of pollinator activities between crop plants and plants within the semi-natural areas [19].

By observing the interactions between plants and pollinators in the community, it is possible to identify the semi-natural plant species that are highly similar to the focal agricultural plant, in regards to their composition of pollinating insects. These semi-natural plant species might be the ones that provide important resources to agriculturally important pollinators, which sustain pollinators across longer time periods. However, plant species might have a high similarity in their composition of pollinating insects by chance if, for example, the plants interact with the most common pollinator species in the community. Null models can be used to distinguish real patterns in similarity from those which are driven by neutral patterns expected from sampling [20,21,22]. These interactions via shared species can be either facultative (by attracting pollinators and leading to an increased chance in conspecific pollen deposition) or competitive (attracting a pollinator away or inhibiting pollination through the deposition of heterospecific pollen). We can measure the potential of one species to indirectly influence another species of the same trophic level based on the frequency of shared interactions (i.e., Müller’s index, [23]).

Bipartite networks that describe observations of plant–pollinator interactions are also an important tool for understanding the community structure and roles of species [24]. Networks are modular in their structure, where species with similar interactions group together, interacting more with each other than with species in different modules [25]. Plants and pollinators often cluster in modules based on their functional traits, due to the important role of trait-matching in determining whether or not species interact. Thus, identifying the plants and pollinators that are important for sustaining the pollination of a focal crop species requires understanding the modular location of the crop, as well as the locations and roles of all other co-flowering plant species in the network. Most species are *peripheral* species; they have links that are almost exclusively with species in their module. Species that are *module hubs* are important for linking species within the module. Species that are *connectors* provide links between modules. Species that are *network hubs* are important within their module and in connecting modules. To ensure the stable and adequate pollination of a focal agricultural species, it is important to have plant species present in the community that are in its module, as well as the connector and network hub species that ensure the cohesiveness of the entire network [26,27].

Oilseed Rape (OSR, *Brassica napus*) is a highly abundant crop in Europe that mass flowers and provides resources for pollinators. Although OSR is self-compatible [28], many studies have found its yields and market value to increase significantly with insect pollination [29,30,31,32]. OSR produces many bright, yellow, entomophilic flowers that secrete high amounts of nectar, making them very attractive to pollinating insects [12,33]. Insect pollination enhances the average crop yield, but overall yield is enhanced by higher visitation rates and not by a higher pollinator richness [7,34]. Many studies have observed the identity of pollinators that provide services to OSR [33,35,36,37] and have examined how the pollinator community is affected by the mass-flowering plant [38,39,40,41,42]. The two studies that have considered OSR in a network context have demonstrated that OSR shares pollinators with plants in hedgerows and surrounding semi-natural grasslands, and attracts some of the most abundant pollinators in the network [39,43].

In this study, we observed plant–pollinator interactions in order to determine the co-flowering plants that are most similar in their visiting pollinator compositions to OSR, and to test if this similarity was higher than expected by chance. We expected to find that OSR attracts abundant pollinators, and therefore it is possible that the similarities in pollinator compositions with many co-flowering plant species are due to chance. Another goal was to quantify the module that OSR is a part of, the identity and functional traits of other plants in that module, and the identity of species that act as connectors and network hubs. We expected that OSR shares a module with co-flowering plants which have similar functional traits, and that it may play a connector or hub role in the network by attracting abundant and generalized pollinator species. Lastly, we used Müller’s index to determine the indirect effect the plants and pollinators have on each other, due to their shared interactions.

## 2. Materials and Methods

Data were collected at six different sites that are 20–35 km away from each other, in Sachsen-Anhalt, Germany (Appendix A), and that are a part of the Terrestrial Environmental Observatories Network (TERENO) [44]. Each site was 4 km × 4 km and was divided into 16 squares of 1 km^2^. From 20 April to 23 May 2017—during the flowering of oilseed rape (OSR, *Brassica napus*)—we used net sampling to collect visiting insects on flowering plants during sunny days, between 9:00 and 15:00 when insects were most active. We sampled within 3–4 squares at each site, in areas that included flowering natural vegetation near OSR fields using a plant-based method, in which an equal amount of time observing pollinators was spent on all flowering plants within 100 m from the field edge, until the sampling saturation was reached (Appendix A). Insects that could be identified in the field (e.g., *Bombus* spp complex, *Apis mellifera*, many Lepidopteran species) were recorded and released. Other insects were collected in vials and labeled with the plant species they were collected from, as well as the site and date of collection. The insects were frozen, pinned, and later identified using published taxonomic guides [45,46,47,48,49,50,51,52,53] and the assistance from a local expert. Insects were identified to a species level when possible, but when it was not, they were identified to genus or family levels. Data were pooled across sites and time periods.

We grouped our plant and pollinator species into functional groups. For plants, we used simplified flower types after Kugler from the BiolFlor database [54], resulting in nine different flower types (Appendix A). We grouped pollinators into eight functional groups based on taxonomic groupings that reflect their life-histories and roles as pollinators (Appendix A). For example, within Hymentopterans, functional groups included honeybees, bumblebees, wild bees, and wasps.

A plant–pollinator network, with all flower visitors and plants, was visualized using the bipartite package in R [22]. We visualized the composition of pollinators on different plant species using nonmetric multidimensional scaling analysis (NMDS), based on Bray-Curtis dissimilarity distances. The NMDS’ were created using the vegan package (function: metaMDS) in R [55]. We tested whether the pollinator composition differed between plants in different functional groups using a permutational multivariate analysis of variance using distance matrices (PERMANOVA), based on the principles of McArdle & Anderson [56].

We created a null model to calculate whether or not the plant species in the semi-natural areas share more pollinator species with OSR than expected by chance. To create the null model, first we calculated the observed Bray-Curtis dissimilarity distance of the pollinator composition for each plant species in the network, as well as for OSR. Bray-Curtis uses a scale from 0 to 1, for which 0 means 100% similarity and 1 means 100% dissimilarity in pollinator community composition. We then randomly assigned pollinators to each plant species based on the observed number of pollinators seen on each plant species and the relative abundances of each pollinator species (i.e., pollinators that were observed frequently were more likely to be chosen). We then re-calculated the Bray-Curtis dissimilarity distance of the pollinator composition for each plant species in the network, and OSR, for this null model. The null model was replicated 1000 times and the mean dissimilarity and its 95% confidence intervals were plotted, along with the observed Bray-Curtis dissimilarity.

Modularity and the modular networks were calculated using the metaComputeModules function in the bipartite package. Modularity is based on a scale from −1 to 1, in which 0 indicates that community division is not better than random, and 1 indicates a strong community structure. We visualized the number of interactions between different functional groups in each module with a bar plot, and tested whether the proportional representation of different functional groups differed between modules using a Chi-squared test. The role of a species—peripherals, module hubs, connectors, or network hubs—can be assigned according to its interactions within its module and within the network. The among-module connectivity (c-value) and within-module degree (z-value) were calculated for both the plants and pollinators in the network using the methods from Olesen and colleagues [57]. Species with low c- and z-values are specialist peripherals, since they have few links within their module and among modules. A connector has a low z- and a high c-value, and are important for connecting several modules together. A module hub has a high z- and a low c-value, and are important for linking species together within its module. A network hub has high z- and high c-values, and are important for the cohesion of the network and within its module. Following the methods from Dormann and colleagues [58], we calculated the 95% quantiles of the c- and z-values using 1000 null models, to objectively set the thresholds for the species roles.

We calculated the Müller Index using the PAC function in R [23]. This index calculates the potential indirect interaction of each plant species to influence all the co-flowering plant species via shared pollinators, and vice versa for pollinators [11,59]. The index is a relative measure and varies between zero (no pollinators/plants shared) to 1 (diet of all visitors depends on the acting plant/visitation to all plants depends on the acting pollinator). A higher value indicates a greater potential for the acting species to influence the target species via shared pollinators for plants, or plants for pollinators. The metric is also asymmetrical, meaning that species A could have more influence on species B than species B on species A.

## 3. Results

Our observed plant–pollinator interaction network consisted of 2778 interactions of 48 plant species and 189 unique pollinators from four orders (Hymenoptera, Diptera, Lepidoptera, and Coleoptera) (Figure 1). The most visited plant was *Brassica napus* (OSR, 26.89% of all visits), followed by *Taraxacum officinale* (23.97%). The functional groups with the most visits were disk flowers with hidden nectar (46.04%), flower heads (25.23%), and lip flowers (14.32%). The most observed pollinator at the species level was *Apis mellifera* (12.99%) which visited 16 (33.33%) different plant species. The most observed functional group of pollinators were wild bees (64 species, 22.53%), followed by bumble bees (nine species, 19.44%) and other fly families (15 families, 17.93%). Percentages for all the species are in Appendix A.

### 3.1. Plant-Pollinator Interactions and Composition

We observed 747 interactions with OSR from 73 different pollinators, 82.2% of which were shared with other plants in the network. The three most frequent visitors were *Apis mellifera*, Mordellidae beetles, and Empididae flies. Over half of the *A. mellifera* observations were on OSR. Likewise, a high percentage of the Mordellidae beetle and Empididae fly observations were on OSR (over 56 and 24% respectively). Unique pollinators visiting OSR, but no other plant species, accounted for only 3.6% of the interactions observed on OSR.

We found that different plant functional groups have significantly different compositions of pollinators (*p* < 0.01) (Figure 2). Disk flowers with hidden nectar were mainly visited by fly species, whereas flower heads were visited by wild bee species, and lip flowers by bumblebees.

OSR had a similar composition of visiting pollinators (based on the Bray-Curtis dissimilarity index) to *Taraxacum officinale*, *Crataegus monogyna*, *Lamium purpureum*, and *L. album*. OSR is a disk flower with hidden nectar and was mainly visited by common, generalist pollinators. *Taraxacum officinale* has yellow flower heads and is pollinated by a wide variety of wild bees and flies. *Crataegus monogyna* is a spring-flowering tree with disk flowers with open nectar and is typically visited by honeybees and beetles. *Lamium album* and *L. purpureum* are white and purple lip flowers that offer nectar to pollinators and are pollinated by many insects, but mainly bumblebees. While all of these species shared many pollinators with OSR, the observed similarity in the composition of visiting pollinators was not significantly higher than that expected by chance, and for some plant species, the pollinator composition was significantly more dissimilar from OSR than expected by chance (Figure 3).

### 3.2. Network Modularity

The network contained 16 modules and a modularity value of 0.47 (Figure 4). OSR was in a module with six other plant species, including *C. monogyna*, and 40 pollinator taxa; the majority of which were honeybees or beetles. *T. officinale* is in a module with two other plant species and forty six pollinators, a majority being wild bees; *L. album*, and *L. purpureum* are in a module with 10 other plant species and 13 pollinators, a majority being bumblebees. The relative abundance of interactions involving different plant and pollinator functional groups significantly differed across modules (*p* < 0.001, Figure 5). The threshold limits for plants were c-value = 0.83 and z-value = 2.27 and for pollinators, c-value = 0.83 and z-value = 2.03. A percentage of 9.28% of all species had an important role in the network (10.40% of plants, 8.99% of pollinators). Three plants were module hubs (*Lamium purpureum*, *Brassica napus* (OSR), *Veronica chamaedrys*) and two plants were connectors (Figure 6a). Empididae flies were a network hub, nine pollinators were modular hubs, and seven were connectors (Figure 6b).

### 3.3. Müller Index

OSR had the highest influence mediated by shared pollinators in the network (Müller index sum = 10.73, mean = 0.22). The species that OSR had a greater effect on were different from those that had a greater effect on OSR (Appendix A). OSR had the greatest effect on *Ranunculus auricomus*, *Adonis vernalis*, *Prunus spinosa*, itself, *Sinapis arvensis*, and *Sorbus aucuparia*, all of which are in the same module. The plants that had the most effect on OSR were *Taraxacum officinale*, *Crataegus monogyna*, *Lamium purpureum*, and *L. album*, all of which shared many pollinators with OSR. The pollinators that had the highest influence on the network were *Apis mellifera* (Müller index sum = 22.2, mean = 0.11), Empididae flies (12.27, 0.06), *Andrena cineraria* (11.74, 0.06), Mordellidae beetles (11.02, 0.06), and *Bombus terrestris* complex (10.06, 0.05).

## 4. Discussion

Our study documents observations of plant–pollinator interactions, revealing that OSR attracts abundant pollinators in the community, and all the similarities in pollinator compositions with co-flowering plants are due to chance. OSR occurs in a module with other disc flowers and plays the role of a module hub, due to the high abundance of pollinators it attracts, which are mostly honeybees, beetles, and flies. OSR has a large influence on the other plants in its module. However, plant species, both in its module and in other modules that also attract abundant pollinators, have the largest influence on OSR. These species include: *Taraxacum officinale, Crataegus monogyna, Lamium purpureum,* and *L. album*. Our results suggest that these plant species provide the resources for pollinators that support the pollination of focal crop species.

We find that the composition of visiting pollinators differs across categories of plants with different functional traits, and that the functional traits of plants and pollinators are clustered into modules in the network. This matches the results of other studies that have found an important role for trait-matching in determining the interactions between plants and pollinators, and the structure of modules [25,60]. Surprisingly, we found that the plant species most similar to OSR in the composition of visiting pollinators were those with dissimilar functional traits that were not members of its module. This is because OSR interacts with common pollinators that are also important to plants in other modules. OSR forms a module with other disk flowers for which honeybee visitors are the most common. However, OSR is also visited by wild bees and flies, which are the predominant visitors in the module that is dominated by plants with flower heads, such as *Taraxacum officinale*. Likewise, OSR is visited by bumblebees, which are the dominant pollinator group in the module that contains lip flower plants, such as *Lamium album* and *L. purpureum*.

We found that OSR acts as a modular hub in the network, and thus is important within its module. By interacting with most of the module’s pollinators (82.5%), OSR ensures stability for the other plants in the module. This is similar to the findings of Stanley and Stout, who found that OSR had a high niche overlap with other plant species in the network [43]. We found in total three module hub plant species, which corresponds with the findings from Dupont that most networks are organized around a few plant hubs [61]. These plant hubs are important for the stability of the network and for supporting a high diversity of plants and pollinators. Loss of these species would fragment the modules and cause the cascading extinction of pollinators.

Our study illustrates the importance of using null models to interpret patterns of pollinator sharing across plant species. In our study, the patterns of similarity in visiting pollinators between OSR and other co-flowering plant species are expected by chance. For example, OSR and *T. officinale* were the most visited plants in the entire network and both were visited by the most abundant pollinators in the system. This finding is in line with another network study on OSR that also found that OSR attracts the most abundant pollinators in the system [39]. Our null model result, combined with the results of other studies that show that the number of visitors, rather than the diversity, influences the reproductive success of OSR [33,62], have implications for management. Specifically, hedgerow plantings and semi-natural areas should be managed in a manner that creates abundant floral resources and habitats to support a high density of pollinating insects. It does not seem necessary to focus on planting co-flowering plant species that share functional traits with OSR.

The plants that OSR affects the most based on the Müller index are those in its module: *Ranunculus auricomus, Adonis vernalis, Prunus spinosa, Sinapis arvensis*, and *Sorbus aucuparia*. We only observed one to three different pollinator species on each of these plants and these pollinators were all common. This suggests that OSR might have a negative effect on these plants by reducing the number of visits these wild plants receive, or lowering the quality of visits to wild plants, if pollinators deliver OSR pollen rather than conspecific pollen. However, a study by Stanly and Stout [43] found that the wild plants that share pollinators with OSR have very little OSR pollen deposited on their stigmas, suggesting that the effects OSR on the pollination of wild plants might be minimal.

Honeybees (*Apis mellifera*) are the most observed flower visitor in our system and play an important role in the network, as seen from their role as a module hub and from their high Müller index. However, less abundant groups of pollinators also have important roles in the network. Other pollinator groups with roles as network hubs and connectors and with a high Müller index included a fly family (Empididae), a beetle family (Mordellidae), and two wild bee species (*Andrena cineraria* and *Bombus terrestris* complex). Empididae flies (dagger flies) are also a known generalist species that thrives in field hedgerows [63] and will expand their foraging breadth to rare plant species when overall plant density is low [64]. Creating habitats that support the nesting, larval, and adult food resources of these pollinators is therefore an important consideration in the management of agroecosystems.

## 5. Conclusions

OSR is a mass flowering crop that plays an important role in plant and pollinator communities during its flowering period. Although it provides abundant floral resources and is a highly attractive plant, this attractiveness could be reducing visitation to co-flowering plants, particularly those in its module. The co-flowering plant species most important to supporting the pollinators of OSR are species that are very common in our region, such as *Taraxacum officinale* and *Lamium purpureum*. These species will naturally colonize semi-natural areas and planted hedgerows and are also found in disturbed areas, such as roadsides and forest margins. Semi-natural areas are important for supporting a high abundance and diversity of insects that provide pollinators’ services to wild and agricultural plant species [16,65]. Studies such as this one, which examined the network structure and sharing of pollinators, contribute to our understanding of the plants and pollinators that are important in agricultural systems. This study identified the wild plants that are most likely to be either facilitated by or compete with OSR, and which ones most influence the pollination of OSR.

## Figures and Tables

**Figure 1 insects-12-01096-f001:**
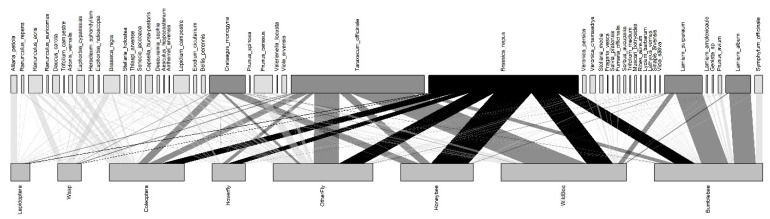
Bipartite network of plant–insect interactions. Plant species are on top and pollinator functional groups on bottom. The thickness of the bars indicates the total number of interactions. OSR and its interactions are black; the other four plants that share a high proportion of interactions with OSR are highlighted in dark grey.

**Figure 2 insects-12-01096-f002:**
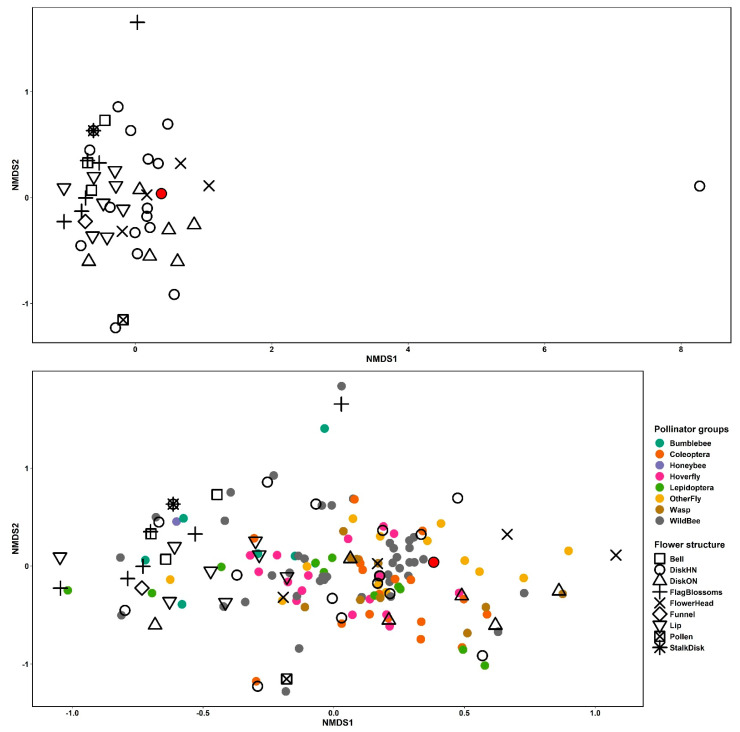
Non-metric multidimensional scaling (NMDS) plot based on the Bray-Curtis dissimilarity distance of insect visitations on plant species (**top**). Bottom is a close-up of ordination with the removal of the outlier, plus the visualization of pollinator species. OSR is highlighted in red. Different functional groups of plants are symbols and pollinator groups are colors. Stress level = 0.13.

**Figure 3 insects-12-01096-f003:**
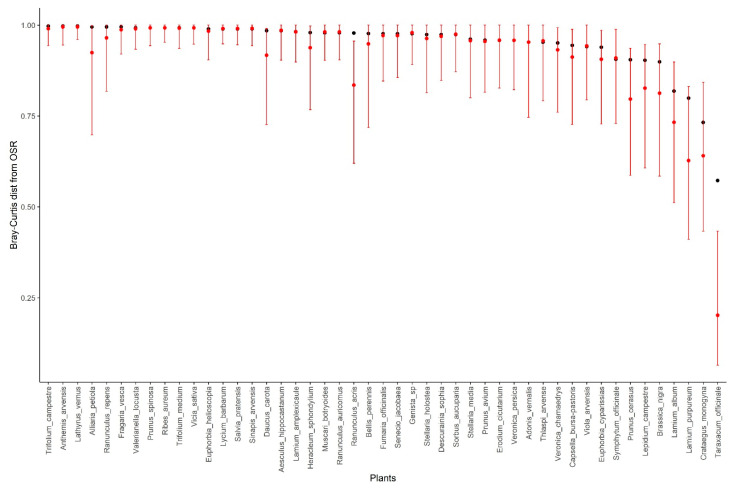
Results of the Bray-Curtis dissimilarity distance of plant species to OSR using a null model. Observed dissimilarity distances are shown in black circles and the mean and 95% confidence intervals of the dissimilarity distances from null models are in red.

**Figure 4 insects-12-01096-f004:**
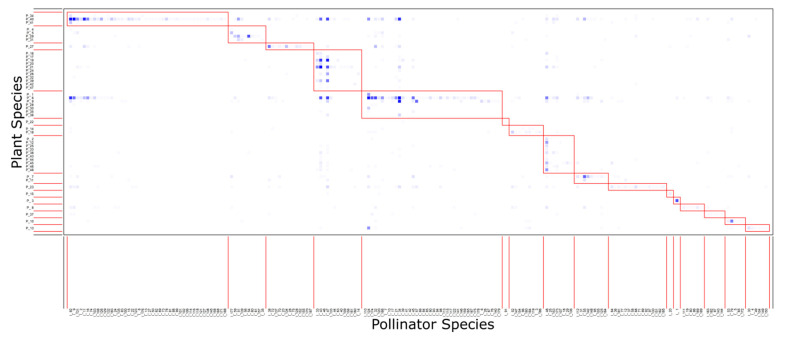
Modular network with 16 different modules. Species are sorted according to their modular affinity, plants as rows and pollinators as columns. Darkers squares indicate more interactions. OSR is in the fifth module. Species names are listed in Appendix A.

**Figure 5 insects-12-01096-f005:**
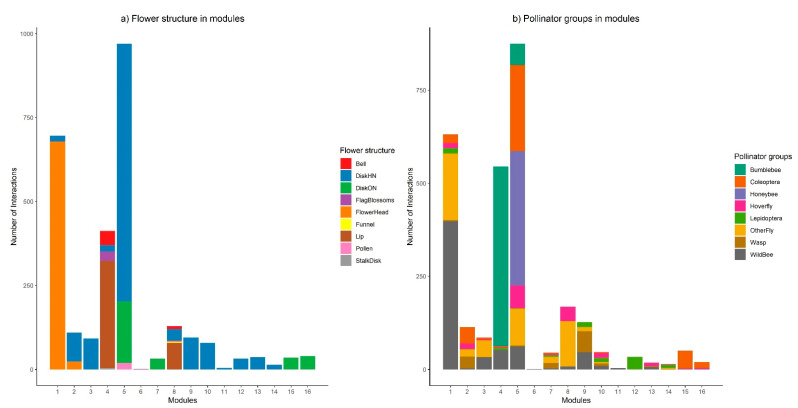
Bar graph showing the abundance of interactions involving different (**a**) plant and (**b**) pollinator functional groups across modules.

**Figure 6 insects-12-01096-f006:**
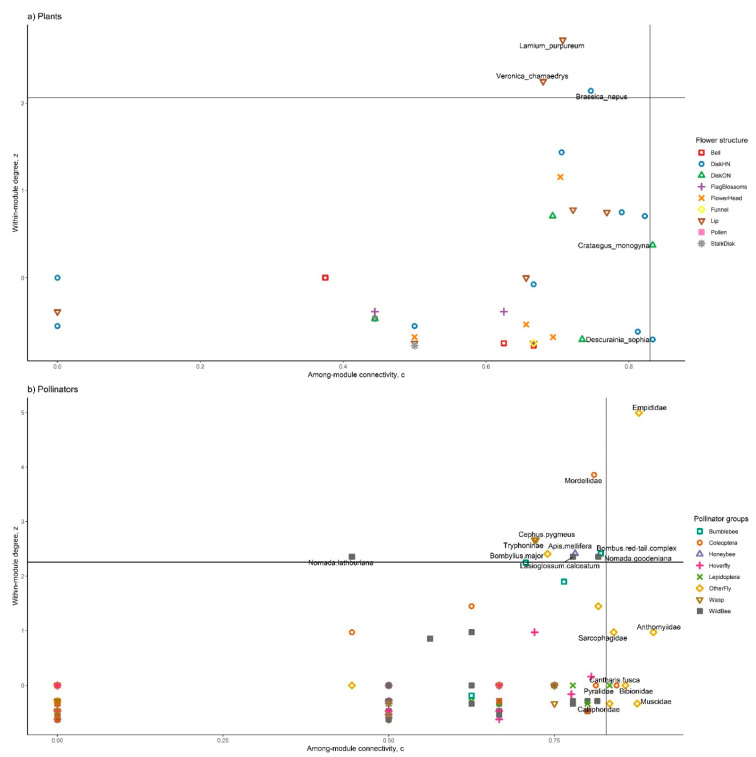
Distribution of (**a**) plants and (**b**) pollinators according to their network role. Each point represents a species, colors and shapes represent functional groups. Species with high c- and/or z-values are named. Threshold lines (95% quartiles) are shown.

## Data Availability

Data and code used in this research can be found on GitHub at https://github.com/AmibethThompson/Oilseed-Rape-Shares-Pollinators. (Accessed on 11 November 2021).

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
