# Peer review of "Oilseed Rape Shares Abundant and Generalized Pollinators with Its Co-Flowering Plant Species"

_insects, 2021, doi:10.3390/insects12121096_

Round 1

Reviewer 1 Report

This study is interesting and although this crop’s pollinator networks have been well documented, the question posed is valid and adds to the understanding of a very important ecological interaction. The manuscript is generally well written and clear but I would like to suggest some concerns below.

The reasoning in the sentence “Networks are modular in their structure, and species within each module interact more with each other than with species in different modules [25]” appears circular and I cannot get an idea what a module would represent ecologically. An example for the network non-expert would help conceptualize this.

The methods describing how sites were sampled is ambiguous as to how the sampling on the OSR fields and the other plant species in non-crop areas were partitioned? The sentence “Plant species and pollinators were sampled to saturation (Table S2)” does not adequately explain how sampling of plants and insects were approached. Normally pollinator surveys use a transect to sample a number of individuals found in a set amount of time. You recorded interactions: each being a pollinator sitting on a plant flower. It is not clear for me how you could get a representative sample of the interactions in a square without giving more information on the floristic composition, especially seeing that square is 1km2! Or did you target all non-crop plant species preferentially? The answers to these questions will have a major effect on the type of interactions you collect.

Specific comments:

The title of the manuscript does not communicate the essence of the results and could be made less generic and more specific.

Throughout the manuscript there are a large number of species names which are not in italics.

Line 41: The sentence states “Wild pollinators can economically contribute on average over $3,000 per hectare towards crop production [8]” and provides a specific value, but no other context is provided to give an idea how it is calculated or how variable this figure is.

Why is Figure 1’s legend in italics?

Figure 4 has no axis labels, but which will help with the interpretation of the figure. Especially seeing as the axis text is so small.

Consider adding more columns to Table S1 to better describe the sampling points than just their gps location. For example, the number of interactions recorded at each site would be useful to describe the sampling effort across the study.

Author Response

Response to Reviewer 1 Comments

This study is interesting and although this crop’s pollinator networks have been well documented, the question posed is valid and adds to the understanding of a very important ecological interaction. The manuscript is generally well written and clear but I would like to suggest some concerns below.

The reasoning in the sentence “Networks are modular in their structure, and species within each module interact more with each other than with species in different modules [25]” appears circular and I cannot get an idea what a module would represent ecologically. An example for the network non-expert would help conceptualize this.

We have clarified the definition of modularity by changing the sentence to “Networks are modular in their structure, where species with similar interactions group together interacting more with each other than with species in different modules [25].”

The methods describing how sites were sampled is ambiguous as to how the sampling on the OSR fields and the other plant species in non-crop areas were partitioned? The sentence “Plant species and pollinators were sampled to saturation (Table S2)” does not adequately explain how sampling of plants and insects were approached. Normally pollinator surveys use a transect to sample a number of individuals found in a set amount of time. You recorded interactions: each being a pollinator sitting on a plant flower. It is not clear for me how you could get a representative sample of the interactions in a square without giving more information on the floristic composition, especially seeing that square is 1km2! Or did you target all non-crop plant species preferentially? The answers to these questions will have a major effect on the type of interactions you collect.

 We have clarified the methods by adding more details. The sentence now reads, “We sampled within 3-4 squares at each site in areas that included flowering natural vegetation near OSR fields using a plant-based method, in which equal time observing pollinators was spent on all flowering plants within 100 meters from the field edge, until sampling saturation was reached.”

Specific comments:

The title of the manuscript does not communicate the essence of the results and could be made less generic and more specific.

We have changed the title to reflect the results better and be more specific. “Oilseed rape shares abundant and generalized pollinators with its co-flowering plant species”

Throughout the manuscript there are a large number of species names which are not in italics.

Thank you for pointing this out. We have gone back through the manuscript and have italicized all species. 

Line 41: The sentence states “Wild pollinators can economically contribute on average over $3,000 per hectare towards crop production [8]” and provides a specific value, but no other context is provided to give an idea how it is calculated or how variable this figure is.

We have simplified the sentence to “Wild pollinators’ economical contribution towards crop production is similar to that of honeybees [8].”

Why is Figure 1’s legend in italics?

Thank you for pointing this out. We have changed it to no longer be italicized.

Figure 4 has no axis labels, but which will help with the interpretation of the figure. Especially seeing as the axis text is so small.

We have added axis labels to the figure as suggested.

Consider adding more columns to Table S1 to better describe the sampling points than just their gps location. For example, the number of interactions recorded at each site would be useful to describe the sampling effort across the study.

Thank you for this suggestion. We have added how many interactions, pollinator species, plant species, and time spent sampled at each location to the table.

Reviewer 2 Report

Title: The attractiveness of oilseed rape in a plant-pollinator network

In this study Thompson et al. studied the pattern of pollinator visitation on oilseed rape and its co-flowering plant species. The work objective is well defined; however, the results and discussion are difficult to understand, perhaps it would be interesting if authors divide the result section into subsections. Moreover, some minor details should be revised (see detailed comments below).

Please, italicize all species names

Line 116: Data were?

Line 129: Data were?

Lines 143: “on the” is repeated

Line 187-188: What percentages do the rest of the species have? A table could be interesting here.

Line 190: the most observed pollinator was Apis, but at what level, species level?

Please explain this part in a clearer way, because, later it is said wild bees as most observed and it is not clear.

Conclusion:  Perhaps it would be interesting if authors made a suggestion of which plants could be interesting in margins of oilseed rape crops?

Line 334-337: revise sentence structure

Supplementary material: remove dots in species names

Author Response

Response to Reviewer 2 Comments

In this study Thompson et al. studied the pattern of pollinator visitation on oilseed rape and its co-flowering plant species. The work objective is well defined; however, the results and discussion are difficult to understand, perhaps it would be interesting if authors divide the result section into subsections. Moreover, some minor details should be revised (see detailed comments below).

 We have added subsections to the results based on our three goals -- Plant-Pollinator Interactions and Composition, Network Modularity, and Müller Index.

Please, italicize all species names

 Thank you for pointing this out. We have gone back through the manuscript and have italicized all species names.

Line 116: Data were?

This change was made as suggested.

Line 129: Data were?

This change was made as suggested.

Lines 143: “on the” is repeated

We have deleted the repeated words.

Line 187-188: What percentages do the rest of the species have? A table could be interesting here.

We have added the percentages for all species in the supplementary table S4 and have added the line, “Percentages for the all species are in Tables S4 and S5.”

Line 190: the most observed pollinator was Apis, but at what level, species level? Please explain this part in a clearer way, because, later it is said wild bees as most observed and it is not clear.

Thank you for pointing this out. We have changed the line to “The most observed pollinator at the species level was Apis mellifera…”

Conclusion:  Perhaps it would be interesting if authors made a suggestion of which plants could be interesting in margins of oilseed rape crops?

We have added these sentences to the conclusion, “The co-flowering plant species most important to supporting pollinators of OSR are species that are very common in our region, such as Taraxacum officinale and Lamium purpureum. These species will naturally colonize semi-natural areas and planted hedgerows and are also found in disturbed areas such as roadsides and forest margins.”

Line 334-337: revise sentence structure

We have rewritten the sentence to, “Studies like this one, which examine network structure and sharing of pollinators, contribute to our understanding of the plants and pollinators that are important in agricultural systems. This study identified the wild plants that are most likely to be either facilitated by or compete with OSR and which ones most influence the pollination of OSR.”

Supplementary material: remove dots in species names

We have made changes as suggested to Table S4 and S5.